# U-Net Performance for Beach Wrack Segmentation: Effects of UAV Camera Bands, Height Measurements, and Spectral Indices

Edvinas Tiškus * , Martynas Bučas , Jonas Gintauskas, Marija Kataržytė and Diana Vaičiūtė

Marine Research Institute, Klaipeda University, 92294 Klaipeda, Lithuania; martynas.bucas@ku.lt (M.B.);
jonas.gintauskas@ku.lt (J.G.); marija.katarzyte@ku.lt (M.K.); diana.vaiciute@ku.lt (D.V.)
* Correspondence: edvinas.tiskus@ku.lt

**Abstract:** This study delves into the application of the U-Net convolutional neural network (CNN) model for beach wrack (BW) segmentation and monitoring in coastal environments using multispectral imagery. Through the utilization of different input configurations, namely, "RGB", "RGB and height", "5 bands", "5 bands and height", and "Band ratio indices", this research provides insights into the optimal dataset combination for the U-Net model. The results indicate promising performance with the "RGB" combination, achieving a moderate Intersection over Union (IoU) of 0.42 for BW and an overall accuracy of IoU = 0.59. However, challenges arise in the segmentation of potential BW, primarily attributed to the dynamics of light in aquatic environments. Factors such as sun glint, wave patterns, and turbidity also influenced model accuracy. Contrary to the hypothesis, integrating all spectral bands did not enhance the model's efficacy, and adding height data acquired from UAVs decreased model precision in both RGB and multispectral scenarios. This study reaffirms the potential of U-Net CNNs for BW detection, emphasizing the suitability of the suggested method for deployment in diverse beach geomorphology, requiring no high-end computing resources, and thereby facilitating more accessible applications in coastal monitoring and management.

**Keywords:** drone; photogrammetry; deep learning; multispectral camera; data combinations





## 1. Introduction

Beach wrack (BW), also known as shore algal deposits or marine debris, is an important component of coastal ecosystems that can provide various ecological, economic, and social benefits [1]. BW is often used as a habitat for a variety of organisms, such as birds and invertebrates, and can serve as a source of food and shelter for these organisms, as well as a source of nutrients for plants [2]. In addition, BW can play a role in protecting the shoreline from erosion and storm waves [3]. It also has economic value, as it can be used as a source of organic matter for soil enhancement and fertilization, and in some cases, can be converted into biogas, a renewable energy source [4]. BW also has cultural and recreational value, as it is often used in traditional practices such as amber collecting and can attract tourists to coastal areas [5]. However, the degradation of BW and the accompanying unpleasant odors may disrupt recreational activities and pose health risks due to the habitation of fecal bacteria, which may thrive in such environments [6].

A complex interplay of meteorological conditions influences the deposition of BW, particularly wave action and storm events. Hydrodynamic measurements have indicated that BW is mostly formed during high sea level and wave events [7]. Furthermore, the morphological evolution of foredunes, which can impact wrack deposition, is driven by wave energy [8]. Storms not only induce deposition but also cause erosion, affecting the equilibrium of beach gradients [9]. These factors collectively contribute to the marine–terrestrial transfer of BW, with significant ecological implications for nearshore environments.

For the monitoring of BW, it is important to understand BW dynamics and the factors that influence its distribution and abundance [10]. However, the detection of BW can be challenging due to its variability in distribution and abundance, its accessibility, particularly in remote or difficult-to-reach areas, and the limitations of traditional methods for mapping it [11,12]. The traditional methods for monitoring BW have been described as labor-intensive and reliant on manual field surveys. A study by Suursaar et al. [7] indicates that BW sampling can be considered a tool for describing the species composition of macrovegetation in near-coastal sea areas. This method involves the physical collection and analysis of BW samples, and while effective, it is subject to human error. Although variable in their applicability, empirical models offer another avenue for monitoring [13]. An integrated framework combining multiple techniques is advocated for comprehensive and effective management [14].

Advanced remote sensing methodologies such as aerial photography, satellite imaging, and light detection and ranging (LiDAR) show potential in identifying BW [15]. Widely recognized spectral indices such as the normalized difference vegetation index (NDVI) and the normalized difference red edge index (NDRE) are pivotal in this domain, exploiting the reflectance attributes inherent to diverse vegetation classes [16]. Furthermore, object-oriented image analysis constitutes another robust strategy to delineate and spatially represent beach zones within the remotely sensed data [17].

According to Yao et al. [18], in many instances, unmanned aerial vehicle (UAV) results outperformed satellite-based techniques. A study by Pan et al. [15] demonstrated that RGB aerial imagery captured with UAVs could be segmented with up to 75% accuracy using machine learning algorithms such as K-nearest neighbor, support vector machine, and random forests. A subsequent study by the same authors employed a camera trap for the continuous monitoring of detached macrophytes deposited along shorelines, offering an efficient and pragmatic method for tracking ecological dynamics [19]. Concurrently, Karstens et al. [20] utilized supervised machine learning methods to map and segment images acquired with UAVs to predict the locations of BW accumulation. Despite these advancements, the studies mentioned limitations, particularly in the number of images utilized for both segmentation and validation, and an imbalanced sample size of classes. While these methods showed promise in terms of their transferability to other areas, they still require additional real-world applications for comprehensive evaluation.

The efficacy of convolutional neural networks (CNNs) in segmenting remote sensing data is contingent on multiple variables, such as the nature and volume of image data. Several limitations to mapping BW should be considered when interpreting the segmentation results. One limitation is the availability and quality of the remote sensing data, which may affect the accuracy and resolution of the BW segmentation. Equally significant is the choice of the CNN model and the accompanying image processing techniques; these parameters directly impact the reliability and accuracy of the results. While the CNN model and image processing techniques are central to achieving high accuracy, the object-specific and environmental variables cannot be overlooked, as they may significantly affect the results' applicability across different locations and times; therefore, a careful selection and optimization of data composition for training and ongoing monitoring are essential for achieving reliable and generalizable outcomes [21]. Research by Lu et al. [22] demonstrated that multispectral images, particularly those with five bands (blue, green, red, red edge, and NIR), yielded accuracy levels comparable to hyperspectral images for vegetation mapping. Concurrently, a study by Wang et al. [23] enhanced landslide detection efficiency by integrating NDVI and near-infrared spectroscopy features, thereby augmenting four-band pre- and post-landslide images to create nine-band composite images.

In the field of remote sensing, digital surface models (DSMs) have shown their utility in complex terrain mapping and analysis, specifically in the context of BW identification and monitoring. For example, Tomasello et al. [24] examined the utility of UAVs for both the volume estimation and segmentation of BW through machine learning techniques, and endorsed this approach for future monitoring initiatives. Moreover, this height information

can be integrated with multispectral imagery captured by UAVs to increase the feature set for the CNN models, thereby enhancing segmentation accuracy for BW mapping.

This study aims to evaluate the U-net model's performance when using six distinct combinations of spectral and height data, to assess the BW area using multispectral imagery from UAV. Additionally, the study aims to compare the performance of this model across different areas of interest (AOIs), by proving the transferability of the model. This research utilizes an extensive dataset, comprising over 150 multispectral 5000-pixel square image tiles. We tested whether the U-Net model's performance in distinguishing BW will not significantly differ across AOIs, thereby demonstrating the model's transferability. We hypothesize that incorporating all available data (multispectral and height) would improve the U-Net model's performance for BW area detection. Also, we tested if the inclusion of height data would have a measurable impact on the final results, contributing to a more comprehensive representation (i.e., volume) of the BW. This study will contribute towards creating a workflow that would not require high-end computing power for CNNs and can facilitate fast, accurate BW estimation without the need for many on-site visits.

## 2. Materials and Methods

### 2.1. Study Area

The study area is located on the exposed coast of the southeastern Baltic Sea (Figure 1). This region is subject to a wind fetch exceeding 200 km, and experiences average wave heights of ~2 m. However, during extreme storm events, wave heights can reach up to 6 m [25]. Four areas of interest (AOIs) were selected along the Lithuanian coastline for monitoring over a year from December 2020 to January 2022. These AOIs represent the four most visited and easily accessible beaches on the continental part of Lithuania. Distinct features, including the proximity to urban areas, the presence of shipping and tourism, dunes, and other coastal features, characterize each of these AOIs (Table 1).

Specifically, Melnrage is located in an area intensively used for shipping and is also close to Klaipeda—the largest city in the western Lithuanian region. Karkle beach is distinguished by the presence of boulders, favorable for the growth of algae, and is far from urban areas, surrounded by many trees; it is the narrowest of the four researched beaches with around 11 m in width [26]. Palanga beach, a popular tourist destination during the summer season, is often cleaned by the municipality, removing larger litter from the sand as well as BW. Sventoji, featuring a fishery port and a popular tourist destination, has the widest sandy beach of all AOIs, measuring around 107 m. All studied beaches have sand dunes, with Karkle beach also featuring clay cliffs. The total length of the beaches in the study area was approximately 39 km, with all coasts exposed to the Baltic Sea.

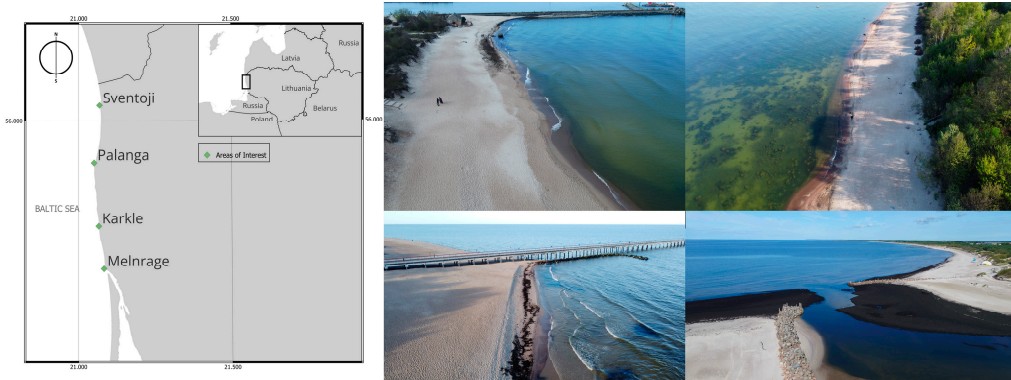

**Figure 1.** Area of interest (AOI) map and images from each one of the four areas, starting from top left and going to bottom right: Melnrage, Karkle, Palanga, Sventoji.

**Table 1.** Description of AOIs according to different attributes.

| Attribute | Melnrage | Karkle | Palanga | Sventoji |
|---|---|---|---|---|
| Proximity to urban area | Close to the port city | Far from urban areas | Close to resort city | Close to resort city |
| Beach cleaning | No | No | Frequently | Frequently |
| Coastal features | Sand dunes | Sand dunes, boulders, and clay cliffs | Sand dunes | Sand dunes |
| Reefs (hard substrate overgrown by macroalgae) | Breakwater | Natural reefs | Natural reefs, groyne, and scaffoldings of pier | Scaffoldings of pier |
| Beach width by Jarmalavičius et al. [26] | ±45 m | ±11 m | ±76 m | ±107 m |

The BW on the Lithuanian Baltic coast is primarily composed (85% of the total relative BW biomass) of perennial red algae (mainly *Furcellaria lumbricalis* and *Vertebrata fucoides*) while filamentous green algae (mainly *Cladophora glomerata*, *C. rupestris*) and brown algae (mainly *Fucus vesiculosus* and *Sphacelaria arctica*), respectively, comprise 14% and 1% of the total relative BW biomass [27]. Red algae species dominate on stony bottoms within depths of 3–16 m, while filamentous green algae densely cover stones in shallower depths (<6 m). Filamentous brown algae such as *Sphacelaria arctica* usually cover hard substrate in deeper parts (>9 m), while overgrowths of *Pylaiella/Ectocarpus* sp. can be found on natural and artificial hard substrates (boulders, piers, scaffoldings) at depths of 1–5 m [28]. Stands of *Fucus vesiculosus* have not been recorded on the hard bottom habitats along the south-eastern Baltic Sea coast, suggesting its transport from other more sheltered coastal areas.

*2.2. UAV-Based Remote Sensing of BW*

A DJI Inspire 2 multirotor UAV equipped with a MicaSense RedEdge-MX multispectral (MicaSense Inc., Seattle, WA, USA) camera was used to acquire the images. The RedEdge-MX camera has 5 bands: Blue (475 nm ± 16 nm), Green (560 nm ± 13 nm), Red (668 nm ± 8 nm), Red edge (717 nm ± 6 nm), and Near-infrared (842 nm ± 28 nm), with 1.2 MP each, and a 47.2° horizontal and 34.4° vertical field of view (micasense.com accessed on 30 October 2023). The RedEdge-MX, with its higher sensitivity (compared to conventional RGB cameras) due to 16-bit image capture, was used for U-Net models. The RedEdge-MX also has additional bands and a global shutter that reduces the risk of blurred images. In addition to multispectral mosaics, RGB mosaics were acquired solely for BW heights, using Zenmuse X5S (DJI, Shenzhen, Guangdong, China) camera (see Section 2.4).

Flights were conducted approximately every 10 days at locations where BW was present and under suitable weather conditions to ensure the quality of the data collected: wind gust speeds of less than 10 m/s, no precipitation, and temperatures above 0 °C (lower temperatures could shorten flight times due to battery performance limitations). If these conditions were not met, the nearest suitable day was chosen for the flight. A flight time was typically scheduled just after sunrise (between 6 a.m. and 10 a.m. local time) to reduce sun glint effects on the water and to minimize the presence of people on the beach, as flights must comply with European regulations prohibiting flying over crowds. The PIX4Dcapture app was used to plan the flights, with a flight height of 60 m. An additional buffer transect was also added to the flight plan to reduce distortions in the center of the final mosaics.

The multispectral camera images had a ground sampling distance (GSD) of ~3.5 cm per pixel, while RGB camera images had a GSD of approximately 1.5 cm per pixel. The mosaics ranged from 0.20 to 1.70 km of beach length, depending on the size of the BW. For U-Net training, 29 multispectral images were mosaiced and partitioned into 163 tiles (Figure 2) of size 5000 × 5000. Out of 75 total flight missions, multispectral images consisted of 7 in Melnrage, 4 in Karkle, 3 in Palanga, and 15 in Sventoji, while the rest were RGB images (see Section 2.4).

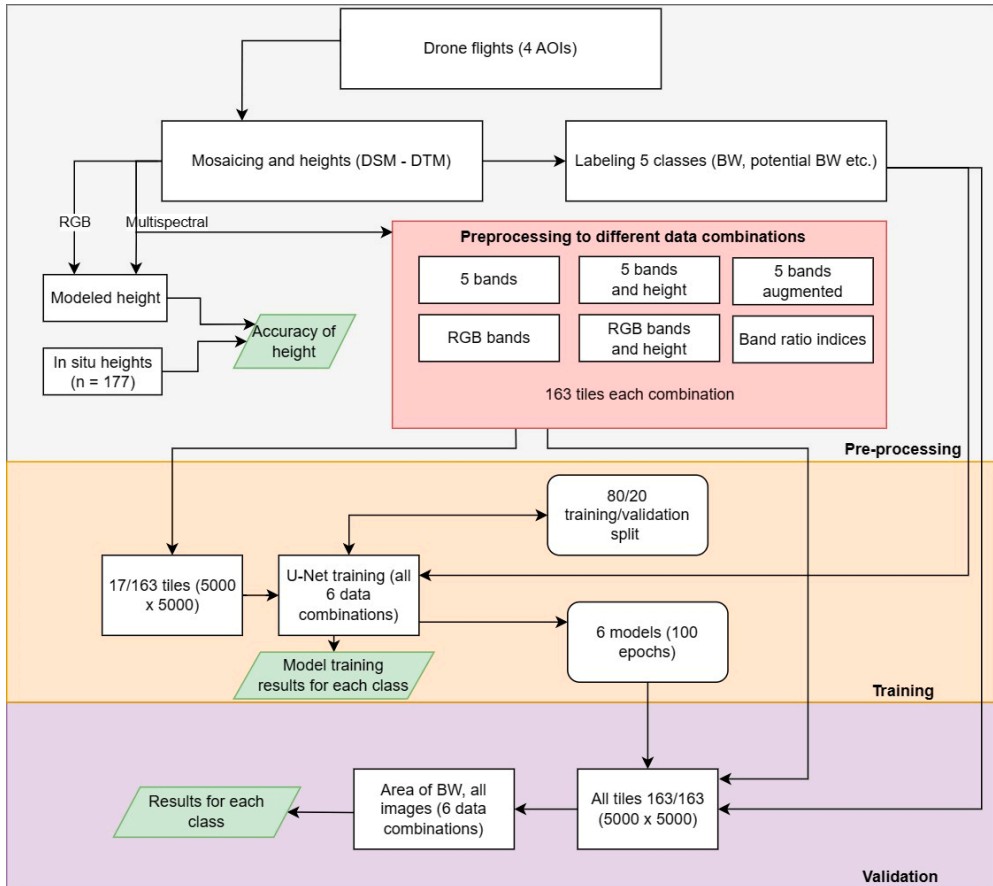

**Figure 2.** Processing workflow for UAV images. Arrows represent image processing from one stage to another. Green squares represent the finished results. Processing workflow for UAV images, including the data augmentation step employing rotations, flips, and other transformations to mitigate spatial location bias and enhance model robustness (see Section 2.3.2).

The PIX4Dmapper 4.6.4 software was used to process the UAV images both from Zenmuse X5S and RedEdge-MX. This software was chosen for its ability to create high-quality image mosaics and generate digital surface models (DSMs) and digital terrain models (DTMs), which are used for calculating the height of BW (see Section 2.4). The mosaics were georeferenced to a Lithuanian orthophoto map with a 0.5 m spatial resolution using QGIS georeferencing tools. At least three ground control points were chosen each time during the georeferencing process, selecting known objects that do not change location, ideally situated in the corners of the final UAV orthophoto.

### 2.3. Machine Learning Methods

#### 2.3.1. Labeling

The multispectral images were mosaiced into three band image files for visual labeling, using the green, blue, and near-infrared bands. The final product of the labeling process is a TIFF file with each pixel assigned to one of five classes: 0 for BW, 1 for potential beach wrack (that is still underwater), 2 for water, 3 for sand, and 4 for other objects (such as buildings, bushes, trees, wooden paths, etc.). It is worth noting that the image background, with a value of Nan, had a large number of pixels in all images, and these were labeled as "other". The labeled images were then opened in ImageJ and exported as TIFF files. Classes were masked by experts, with the main goal of marking the areas of BW accumulations. In some cases, the labeling was done roughly, where BW was spread out in many pieces at a small scale (Figure 3).

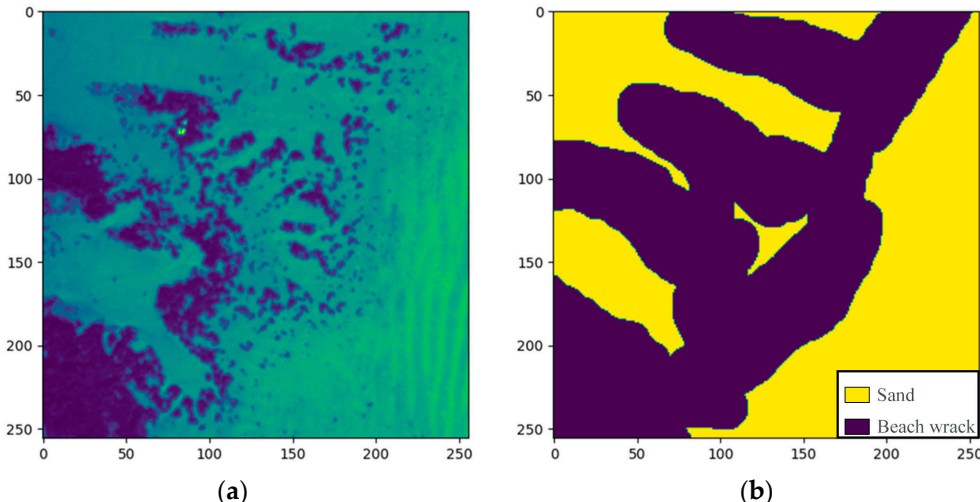

**Figure 3.** Example of manual labeling and its rough mask of BW in some areas at a pixel level, where (**a**) is a single red band with color pallet and (**b**) are the labeled areas of the same image. X and y coordinates show the locations of pixels (256 × 256) equal to around 8 m$^2$.

The accurate labeling of the mosaic tiles allows the U-Net CNN model to distinguish BW from other classes in the scene, such as sand, water, or other objects. It provides data against which the model's predictions are evaluated, enabling the assessment of its effectiveness in BW identification and quantification. Labeling was performed on orthomosaic tiles with a maximum size of 5000 by 5000 using the "Labkit" [29] plugin in ImageJ FIJI. This plugin uses traditional supervised machine learning to assist with labeling using given samples, which were manually reviewed, and any incorrect labels were corrected by an expert. The near-infrared band was particularly useful in distinguishing between small rocks and BW, which can be challenging to differentiate in RGB images, as BW consists of algae that have chlorophyll-a, which is more reflective in the near-infrared band spectrum.

### 2.3.2. Data Pre-Processing

The model training was performed on a computer equipped with 32 GB RAM, an Intel Core i7 8th gen (Intel Corporation, Santa Clara, CA, USA) CPU, and an NVIDIA GTX 1070 (NVIDIA Corporation, Santa Clara, CA, USA) GPU (8 GB vRAM). To accommodate the memory constraints inherent to deep learning approaches, high-resolution tiles were partitioned into smaller 256 × 256 pixel segments. These reduced dimensions were sufficient to maintain the visibility of the objects relevant to the study's context.

Out of 163 tiles generated from the partitioning, 17 were selected by expert judgment for inclusion in the model training set (Table 2). The selection aimed to include at least one tile from each date and AOI, to ensure a comprehensive representation of all segmentation classes.

For basic image manipulation (merging, selecting bands, augmentation processes, etc.), Python with GDAL 3.4.3 [30] library was used. Six different combinations from multispectral data were used to train the final models to assess the impact of different data types on the model's performance. The combinations included the use of RGB bands, RGB and heights, 5 bands, 5 bands and height, augmented, and the band ratio indices merged into one TIFF, and will each be detailed later in this section to explain their combination process.

**Table 2.** The partitioning of training data for the U-Net CNN model. Images corresponding to each AOI and date. Check marks (✓) indicate tile of AOI and data and multiple check marks (✓✓) show that multiple tiles were used from the same date and AOI.

| Date/AOI | Melnrage | Karkle | Palanga | Sventoji |
|---|---|---|---|---|
| 25 August 2021 | ✓ | | | ✓✓ |
| 8 September 2021 | | | | ✓ |
| 15 September 2021 | ✓ | | ✓ | |
| 17 September 2021 | | ✓✓ | | ✓ |
| 22 September 2021 | ✓ | | | ✓ |
| 29 September 2021 | | | | ✓ |
| 1 October 2021 | | | | ✓ |
| 26 October 2021 | | ✓ | | ✓ |
| 4 March 2022 | ✓ | | | |
| 22 March 2022 | | | | ✓ |

The indices included the normalized vegetation index (NDVI) (1), the normalized difference water index (NDWI) (2), and the normalized difference red edge index (NDRE) (3):

$$NDVI = \frac{NIR - Red}{NIR + Red} \tag{1}$$

$$NDWI = \frac{Green - NIR}{Green + NIR} \tag{2}$$

$$NDRE = \frac{NIR - Red\ edge}{NIR + Red\ edge} \tag{3}$$

where each remote sensing reflectance (Rrs) band is represented by a band name.

The choice of NDVI, NDWI, and NDRE over other indices was based on their specific spectral sensitivities relevant to BW identification. NDVI leverages red and NIR spectral bands, which are well established in vegetation studies, offer robust data on plant health [31,32], and are directly relevant to BW mapping, as it mostly consists of macroalgae. NDWI, which computes reflectance from the green and NIR spectral regions, helps distinguish water and land areas, and is useful in detecting potential underwater BW. NDWI is important in delineating water features and is crucial for identifying submerged or partially submerged vegetation [33,34]. However, NDWI may be impacted by shadows and surface roughness, necessitating its use alongside other indices. Lastly, the NDRE index helps to measure the amount of chlorophyll-a in the plants, and it can also be used for biomass estimation [35], which is also related to BW and the amount of it.

Data augmentation was undertaken as an exploratory measure to investigate potential spatial location bias related to class pixel locations within the dataset, rather than as a strategy for genuine model improvement. It was implemented solely on a single dataset that incorporated all spectral bands and the heights (see Section 2.4). Data augmentation was implemented by manipulating images through specific transformations: random rotations of images at defined angles (0°, 90°, 180°, and 270°), and horizontal and vertical flips, each with an equal probability of 50%. This methodological approach ensures a diverse dataset, enhancing the robustness of the subsequent analyses.

### 2.3.3. U-Net Semantic Segmentation

The U-Net architecture, introduced by Ronneberger et al. [36], was selected for this study due to its precision in localization and its ability to effectively handle smaller datasets for complex image segmentation tasks. The distinguishing attribute of CNNs lies in their capacity to master spatial feature hierarchies, effected through the use of convolutional strata that scrutinize the input image, consequently deploying filters to abstract features across various scales. In this paper, a similar architecture (Figure 4) was used to the one described in the original U-Net paper, with the addition of extra layers for the multispectral

images and a reduced input image size. Also, padding and a dropout of 20% was used, which is a regularization technique that involves randomly dropping a certain percentage of the neurons in the model during training, which helps to prevent the model from becoming too complex and overfitting the training data [37].

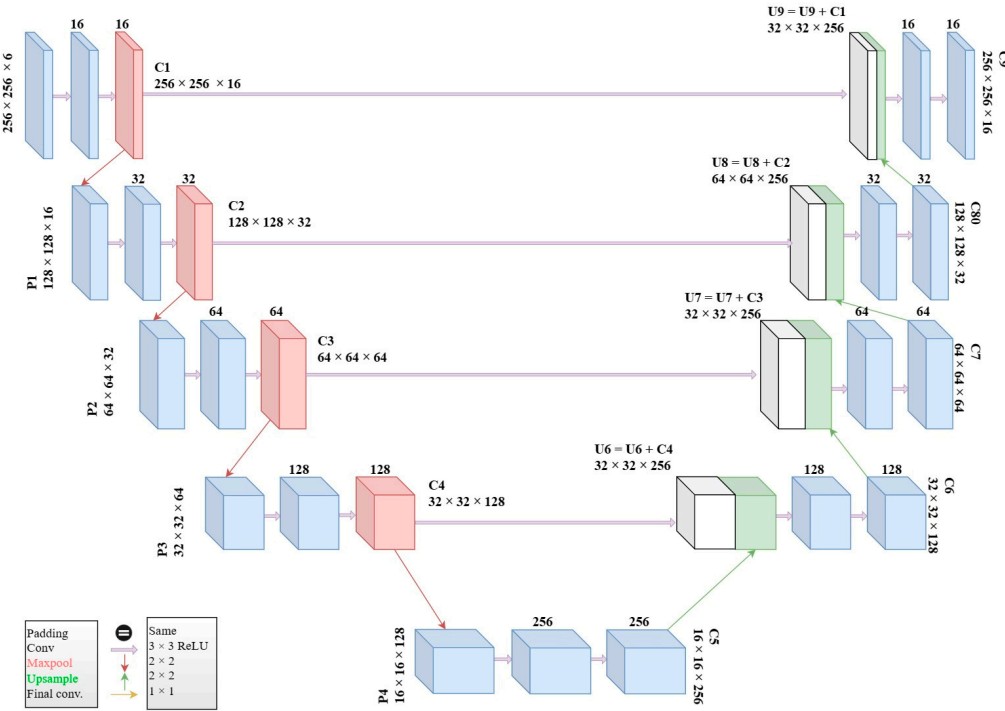

**Figure 4.** U-Net architecture (modified from Ronneberger et al. [36]).

The training itself was conducted in Python 3.9 using Keras version 2.3.1 [38] for model construction, with custom operations implemented in TensorFlow 2.1.0 [39]. The U-Net model was trained using a batch size of 16 patches (i.e., in each iteration of an epoch, 16 images were processed together), as it was the maximum limit for the computing power used in this study. The training was set to run for 100 epochs, but an early stopping mechanism was implemented to prevent overfitting. The training was halted if the model's performance did not improve after 6 consecutive epochs. This approach ensured that the model was not overtrained on the data, which could lead to a poor generalization of the testing data. The training models showed that all datasets around the first 20 epochs' results improved the most (Figure 5) for validation and training loss. After the 20th epoch, training and validation loss still decreased, but at a slower pace, while validation loss did not improve near epoch 40.

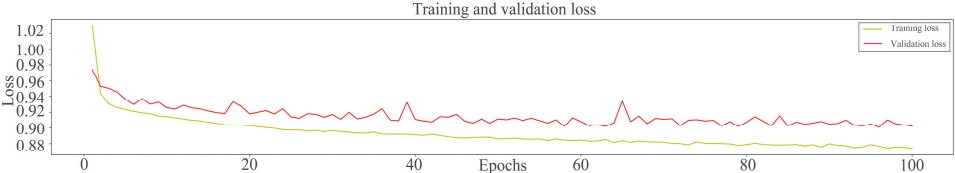

**Figure 5.** Example of loss for training and validation over 100 epochs. The dataset used for this training was all 5 bands and height.

The workflow for image segmentation began by assigning labeled TIFFs to the final pre-processed images. All classes were given equal weight, and the loss function was defined as the combination of dice loss and focal loss. The dice loss measure [40] quantifies the overlap between classes on a scale from 0 to 1, with higher values indicating better performance. The focal loss [41] helps to address the issue of unbalanced class

distributions by decreasing the contribution of well-trained pixels and focusing on poorly trained ones.

To eliminate the edge effect when patching images, the Smoothly-Blend-Image-Patches [42] package was used, which employs a U-Net for image segmentation and blends predicted patches smoothly through 2D interpolation between overlapping patches.

### 2.4. BW Heights

In addition to multispectral mosaics, 16 RGB mosaics were acquired for assessment of BW heights in Melnrage, 11 in Karkle, 6 in Palanga, and 13 in Sventoji using the Zenmuse X5S RGB camera that has an RGB lens with 20.8 MP and a 72° field of view (dji.com accessed on 30 October 2023).

To validate the UAV-derived height of BW deposits, a total of 16 in situ sampling missions were carried out concurrently with UAV flights (Table 3). The height of BW deposits was initially assessed using a plastic ruler at the study site. To ensure accuracy, the ruler was placed gently on the deposits to prevent penetration into the underlying sand, and was aligned vertically to measure at around every 10 m of BW, in a transect line of three points: the start of the BW (near the water), middle point selected by expert judgement, and the end of the BW (furthest from the water). They comprised a total of 177 points within each site, covering areas of BW deposits and reference areas without BW.

**Table 3.** In situ sampling of BW on the coast and in the water at four study sites from December 2020 to January 2022. Bolded dates indicated when the RGB camera was used and not bolded when the multispectral camera was used. The number of height measurements per sampling is provided in brackets.

| Melnrage | Karkle | Palanga | Sventoji |
|---|---|---|---|
| **2021.04.20 (3)** | **220.12.05 (1)** | **2020.12.05 (2)** | **2020.12.05 (4)** |
| **2021.06.02 (20)** | **2021.07.27 (3)** | **2021.07.29 (3)** | **2021.07.07 (10)** |
| **2021.06.18 (11)** | 2021.09.17 (23) | | 2021.08.27 (3) |
| **2021.08.10 (8)** | | | 2021.09.17 (58) |
| 2021.09.16 (25) | | | |
| 2022.01.24 (3) | | | |

The estimation of the BW height from the UAV images involved subtracting the DSM from the DTM using GDAL.

### 2.5. Performance Metrics

To validate the model's performance during training, the data were randomly split into two sets, 80% for training and 20% for validation, according to common practice to avoid overfitting and test the model's ability to generalize. This split ensured that the model was trained on a large enough dataset to learn the necessary features, while also having a separate set of data to test its performance [43]. A separate validation set, consisting of all tiles, was used to assess the model's ability to generalize to new data and ensure that it was not overfitting to the training data.

Several metrics were employed to assess the model's performance: precision, recall, F1 score, and Intersection over Union (IoU). Precision quantifies the proportion of correctly predicted positive values to the total predicted positives, while recall measures the fraction of correctly predicted positive values to the total actual positive values. The F1 score harmoniously combines precision and recall, providing a balanced performance metric [44]. The IoU, also known as the Jaccard index, offers a comprehensive assessment of the model's performance, going beyond pixel accuracy to measure the similarity between the predicted and ground truth labels [45]. In general, models trained on specific datasets will have a higher IoU than models trained to be more general, but the latter will have a wider range of applicability [46]. The effectiveness of the selected models was evaluated on testing data by comparing the IoU metric. The IoU was also compared for each AOI and each class. No

single IoU threshold fits all use cases; however, it is common practice to use a threshold of 0.5 for accurate segmentation [47]. Therefore, IoU values above 0.7 were considered as high, from 0.5 to 0.7 as moderate, and below 0.5 as low.

In addition, the IoU between labeled and segmented BW for tiles in the whole mosaic BW areas were calculated and compared with each other as well. Furthermore, for the comparison of IoU between AOIs, the normality and homogeneity of variance assumptions were tested, using the Shapiro–Wilk and Levene's tests, respectively. Given the violations of normality and homogeneity of variance assumptions, the Dunn's test post hoc pairwise comparisons of IoU between the AOIs was utilized. The *p*-values were adjusted using the Bonferroni correction to control for multiple comparisons. The comparison between averages was performed with a one-way ANOVA test. All statistical analyses were performed using numpy [48], scipy [49], statsmodels [50], and sklearn [51] Python packages, at a significance level of 0.05.

In situ measured heights and heights calculated from UAV were assessed for correspondence using Pearson's correlation coefficient (r). The precision of these measurements was further quantified by the root mean square error (RMSE) and mean absolute error (MAE). This was also tested for separate AOIs.

## 3. Results

### 3.1. Performance of Various Input Training Data

In training the U-Net model's performance across various data combinations, the "band ratio indices" combination consistently showcased the best results (Table 4), especially for the segmentation of BW. With this combination, the model achieved an F1 score of 0.86 and an IoU of 0.75 for BW. Notably, the "5 bands" combination also delivered good results, particularly for potential BW, with an F1 score of 0.57 and an IoU of 0.40. However, when examining the potential BW class, all combinations presented relatively lower IoU scores. The "augmented data" combination displayed the least promising outcomes across the metrics.

**Table 4.** IoU, precision, recall, and F1 scores for different classes resulting from a convolutional neural network U-Net model's training set, on various data combinations. The columns in the table represent different datasets, while the rows contain the performance scores for each class. These results were obtained after 100 epochs of training. Best performing values for average, BW, and potential BW are marked with the * symbol.

| Dataset Type | 5 Bands and Height | 5 Bands | RGB | RGB and Height | Augmented Data | Band Ratio Indices |
|---|---|---|---|---|---|---|
| IoU avg. | 0.67 | 0.71 * | 0.69 | 0.69 | 0.66 | 0.67 |
| Beach wrack | 0.72 | 0.73 | 0.71 | 0.66 | 0.67 | 0.75 * |
| Potential beach wrack | 0.35 | 0.4 | 0.35 | 0.38 | 0.3 | 0.39 * |
| Water | 0.68 | 0.73 | 0.69 | 0.73 | 0.7 | 0.65 |
| Sand | 0.75 | 0.81 | 0.76 | 0.78 | 0.74 | 0.71 |
| Other | 0.86 | 0.89 | 0.93 | 0.92 | 0.88 | 0.86 |
| F1 score avg. | 0.87 | 0.9 * | 0.88 | 0.89 | 0.87 | 0.86 |
| Beach wrack | 0.83 | 0.84 | 0.83 | 0.79 | 0.8 | 0.86 * |
| Potential beach wrack | 0.52 | 0.57 * | 0.51 | 0.55 | 0.46 | 0.56 |
| Water | 0.81 | 0.85 | 0.82 | 0.84 | 0.83 | 0.79 |
| Sand | 0.86 | 0.89 | 0.86 | 0.88 | 0.85 | 0.83 |
| Other | 0.94 | 0.96 | 0.97 | 0.97 | 0.96 | 0.94 |
| Precision avg. | 0.88 | 0.90 * | 0.89 | 0.9 * | 0.88 | 0.87 |
| Beach wrack | 0.76 | 0.87 | 0.87 | 0.89 * | 0.83 | 0.79 |
| Potential beach wrack | 0.51 | 0.54 | 0.5 | 0.48 | 0.37 | 0.8 * |
| Water | 0.77 | 0.83 | 0.79 | 0.82 | 0.79 | 0.77 |

**Table 4.** *Cont.*

| Dataset Type | 5 Bands and Height | 5 Bands | RGB | RGB and Height | Augmented Data | Band Ratio Indices |
|---|---|---|---|---|---|---|
| Sand | 0.87 | 0.89 | 0.88 | 0.91 | 0.89 | 0.79 |
| Other | 0.99 | 0.98 | 0.98 | 0.97 | 0.98 | 0.98 |
| Recall avg. | 0.87 | 0.89 * | 0.88 | 0.89 * | 0.87 | 0.86 |
| Beach wrack | 0.93 | 0.81 | 0.79 | 0.72 | 0.77 | 0.94 * |
| Potential beach wrack | 0.53 | 0.6 | 0.53 | 0.66 * | 0.58 | 0.43 |
| Water | 0.86 | 0.87 | 0.86 | 0.86 | 0.87 | 0.8 |
| Sand | 0.85 | 0.9 | 0.84 | 0.85 | 0.82 | 0.88 |
| Other | 0.9 | 0.95 | 0.97 | 0.97 | 0.93 | 0.91 |

The post hoc test revealed that none of the pairwise comparisons were statistically significant ($p \geq 0.74$), suggesting that different data combinations did not significantly impact the IoU scores.

The "5 bands" combination yielded the best results for the sand and water classes, achieving the highest F1 scores and IoU values among the combinations. In contrast, the "RGB" combination was the most effective for the other class, showcasing exemplary F1 scores and IoU values. The precision and recall rates for each of these optimal combinations were also notably high, confirming the findings.

### 3.2. Validation of Trained U-Net Model for Testing Data

In the segmentation of BW, the combination that used "RGB" bands yielded the best performance with an IoU of 0.42 (Figure 6) and further demonstrated an F1 score of 0.54. Following closely, the combination utilizing "augmented data" had an IoU of 0.41, supported by an F1 score of 0.55. The "5 bands and height" combination also showcased notable performance with an IoU of 0.39 and an F1 score of 0.54. Conversely to training data, for validation the "band ratio indices" combination yielded the lowest IoU of 0.37 for BW classification, alongside an F1 score of 0.50.

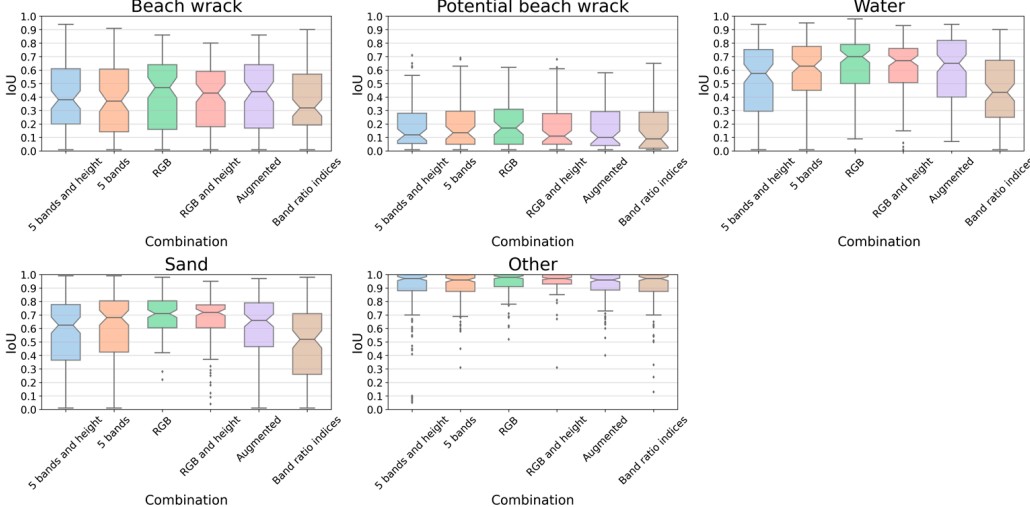

**Figure 6.** The boxplots present the IoU scores for the six different data combinations applied during the U-Net (CNN) model validation. The plots show the distribution of IoU scores for each segmentation class: BW, potential BW, water, sand, and other. The central line inside each box represents the median, while the edges of the box indicate the 25th and 75th percentiles. Outliers may be represented by individual points.

The "5 bands and height" combination emerged as the most effective for potential BW segmentation, recording an IoU of 0.20 and 0.38 for the F1 score. The "RGB" and "5 bands" combinations followed closely, with an IoU of 0.20. While the "RGB" combination achieved an F1 score of 0.46, the "5 bands" combination had an F1 score of 0.38. The "augmented data" combination exhibited the least efficacy in segmenting potential BW, with the lowest IoU of 0.16 and accompanying F1 score of 0.34.

Regarding the additional classes, in the water class, the "RGB" combination emerged superior with an IoU of 0.64 and an F1 score of 0.76. In contrast, the "band ratio indices" combination exhibited the lowest performance, securing an IoU of 0.45 and an F1 score of 0.58. In the sand class, the "RGB" combination outperformed the rest with an IoU of 0.70 and 0.82 for the F1 score, while the "band ratio indices" combination trailed with an IoU of 0.48, alongside an F1 score of 0.61. For the class of other, the "RGB and height" combination achieved the highest IoU of 0.95, supported by an F1 score of 0.97, whereas the "5 bands and height" combination had the lowest IoU of 0.87, with an F1 score of 0.91.

For the overall average performance of all combinations, there was no significant difference between them (f = 0.10, *p* > 0.05). The "5 bands" combination achieved an F1 score and IoU of 0.88 and 0.54, respectively. When height was incorporated, the "5 bands and height" combination demonstrated a slight dip in performance, with average metrics for the F1 score at 0.85 and an IoU of 0.51. The "augmented data" combination showcased metrics closely resembling the "5 bands" combination, with 0.88 for F1 score and 0.54 for IoU. A noticeable decrease in average performance was observed with the "band ratio indices" combination, yielding 0.84 and 0.47 for the F1 score and IoU, respectively. The "RGB" combination recorded the highest average metrics among all combinations: F1 score of 0.92 and IoU of 0.58. Lastly, the "RGB and height" combination mirrored the "RGB" combination in precision and recall, but displayed a slightly lower average F1 score and IoU of 0.92 and 0.57, respectively.

Comparing the segmentation results of BW between AOIs, Dunn's post hoc tests for IoU showed significant differences between Karkle and the rest of the AOIs (*p* < 0.05), while no significant differences (*p* > 0.05) were observed between Melnrage, Palanga, and Sventoji (Figure 7).

Specifically, in Sventoji, the "5 bands and height" combination yielded the highest IoU at 0.48 ± 0.26, while in Palanga, the "RGB and height" combination was most effective with an IoU of 0.46 ± 0.22. For the class of potential BW, the "RGB and height" combination in Karkle registered an IoU of (0.29 ± 0.22), and in Melnrage, the "RGB" combination yielded (0.26 ± 0.19).

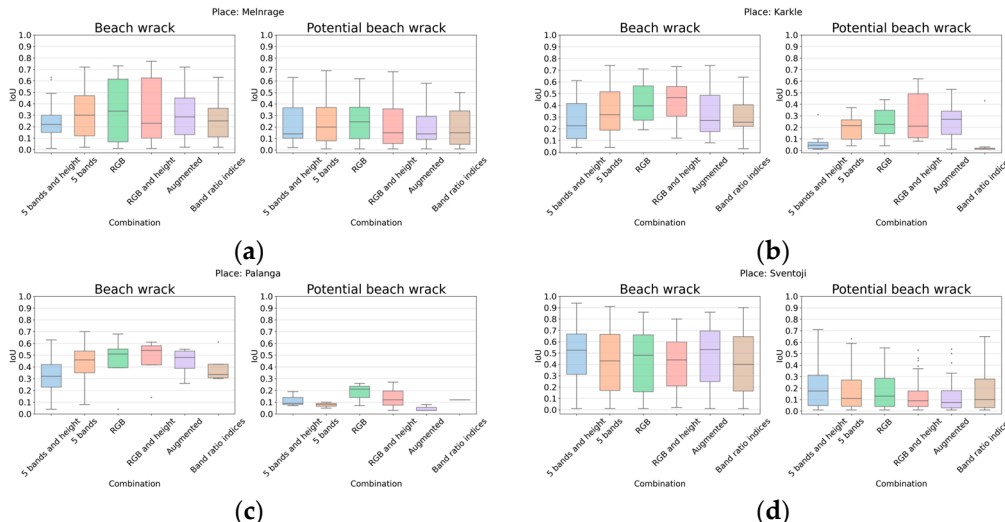

**Figure 7.** Boxplots for each AOI separately where (**a**) is Melnrage, (**b**) Karkle, (**c**) Palanga, and (**d**) Sventoji. Each boxplot represents the results for all data combinations, and notches show a confidence interval around the median.

For the water class, the "RGB" combination in Melnrage produced an IoU of (0.63 ± 0.23), followed by the "RGB and height" combination in Karkle with (0.50 ± 0.19). In the sand class, the "RGB and height" combination in Karkle led with an IoU of (0.65 ± 0.25), closely followed by the "RGB" combination in Melnrage, having an IoU of 0.68 ± 0.15. Lastly, for the other class, the "RGB and height" combination in Karkle achieved the highest IoU at (0.93 ± 0.06), while Melnrage scored (0.94 ± 0.09) using the "RGB" combination.

### 3.3. Heights and Areas of BW

The labeled areas of BW were from approximately 235.55 m$^2$ to 11193.33 m$^2$, while the area of BW derived from the U-Net model using the "RGB" combination exhibited a wider range, from 8.83 m$^2$ to 3710.01 m$^2$ (Figure 8). While the relationship was generally linear between the labeled BW areas and areas retrieved using the U-Net model with the "RGB" combination, there was a relatively large average with standard deviation, namely, a labeled area of 1887.94 ± 2198.93 m$^2$, corresponding to the area of 1217.80 ± 939.90 m$^2$ derived from the U-Net model using the "RGB" combination.

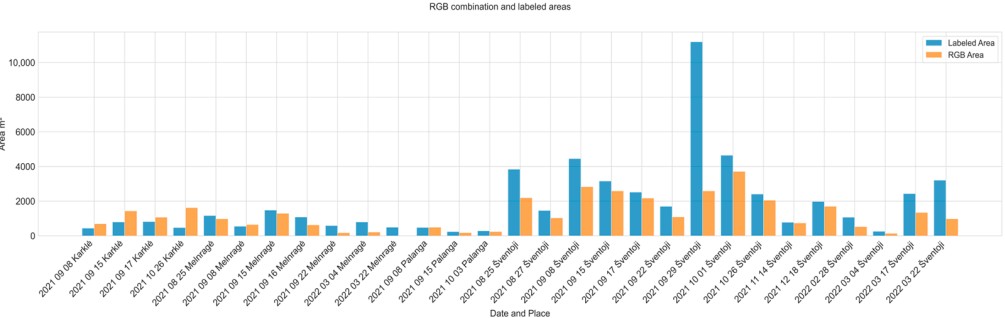

**Figure 8.** The areas of BW coverage in the investigated AOIs retrieved from UAV after the application of the U-Net model with the "RGB" combination and labeled BW areas.

Palanga had the best agreement comparing labeled to RGB areas, with an average of 39.09 ± 39.43 m$^2$. For Karkle, all areas were overestimated with an average of −572.05 ± 427.17 m$^2$. As for Sventoji, it had the largest average, 3005.83 ± 2603.98 m$^2$ of BW area, and the differences were also the largest, 1295.03 ± 2118.10 m$^2$. In Melnrage, most of the values were underestimated except for one on 8 September 2021, and the average overestimation was 315.66 ± 238.01 m$^2$.

While comparing labeled to segmented areas of BW, the "RGB" combination exhibited the highest correlation coefficient (r = 0.87) among all tested approaches for agreement with the area, followed closely by the "RGB and heights" combination with an r of 0.86. Additionally, both these models had the lowest MAE and RMSE values, 562.27 and 783.59 for "RGB", and 658.28 and 897.08 for "RGB and height", respectively.

Other data combinations (Table 5) had lower correlation coefficients ranging from 0.46 for "5 bands" to 0.73 for "augmented data" combinations. The MAE and RMSE were also worst for "5 bands" at 825.54 and 1377.34, respectively, and for the "augmented data" combination, that was the next best combination after "RGB" and "RGB and height", with a MAE of 575.91 and an RMSE of 902.87.

The average calculated height of BW (0.46 ± 0.40 m) from UAV overestimated the in situ measured height by five-fold (0.09 ± 0.11 m) from a sample size of 177 (Figure 9). The maximum BW height calculated was 2.37 m, while the maximum in situ measurement was only 0.52 m, with a standard deviation of calculated height—0.03 and in situ—0.01 m. The correlation between modeled and in situ heights was 0.44 ($p < 0.05$).

**Table 5.** Statistics between labeled and segmented areas of BW. Pearson's correlation coefficient—r, MAE—mean absolute error, RMSE—root mean square error.

| Data Combinations | r | MAE | RMSE |
|---|---|---|---|
| 5 bands and height area | 0.48 | 807.99 | 1512.91 |
| Augmented data area | 0.73 | 575.91 | 902.87 |
| Band ratio indices area | 0.68 | 648.42 | 1097.48 |
| 5 bands area | 0.46 | 825.54 | 1377.34 |
| RGB area | 0.87 | 562.27 | 783.59 |
| RGB and height area | 0.86 | 658.28 | 897.08 |

From the example of the visual representation of all AOIs (Figure 10), it is evident that the model's performance is adequate in accurately classifying the majority of the BW. In these examples, Melnrage is overestimated by 455.10 m$^2$, Karkle underestimated by 251.59 m$^2$, Palanga overestimated by 56.75 m$^2$, and Sventoji overestimated by 934.70 m$^2$.

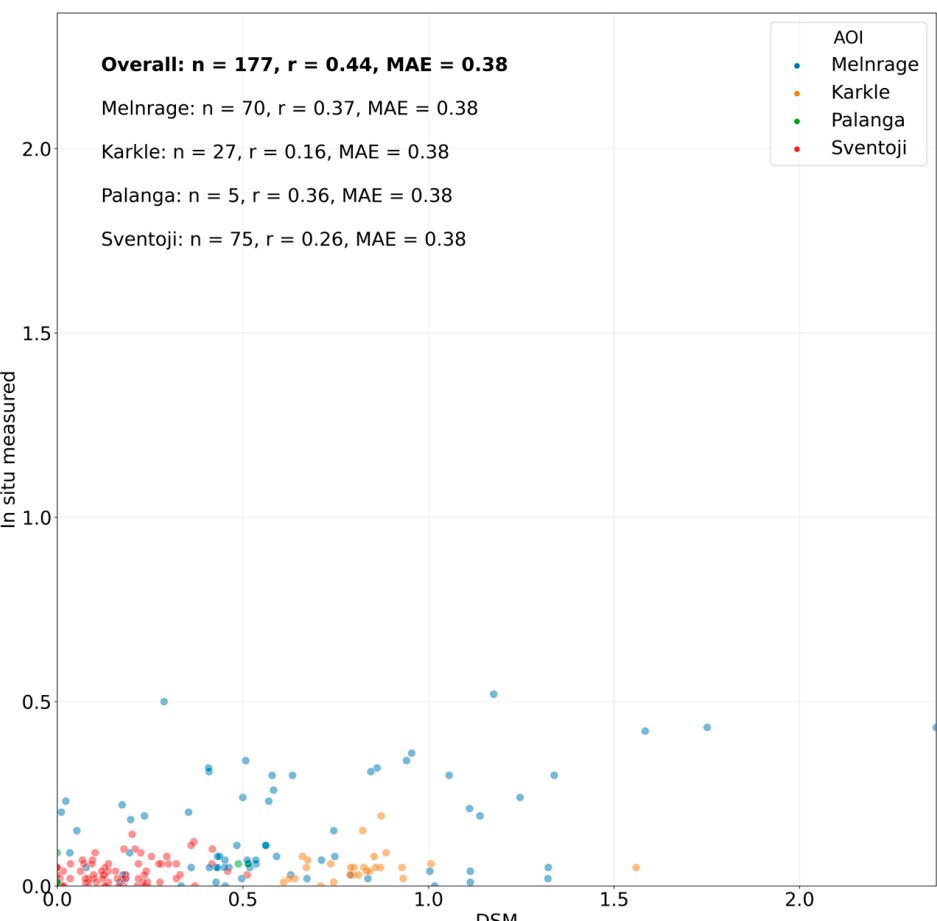

**Figure 9.** Agreement between in situ height and mosaic-calculated height. Different colors represent different AOI. r—Pearson's correlation coefficient, MAE—mean absolute error.

This precision captures the expected locations and distribution patterns of all classes, confirming the model's robustness. Specific regions, such as Palanga and Melnrage, present minor challenges, with a few discrepancies in detecting the potential BW. However, these instances are more the exception than the norm. The sand and water classes have the best visual results with few minor variations. Similarly, the class of other is also excellent, with just a few objects, mainly in Palanga, misclassified as sand.

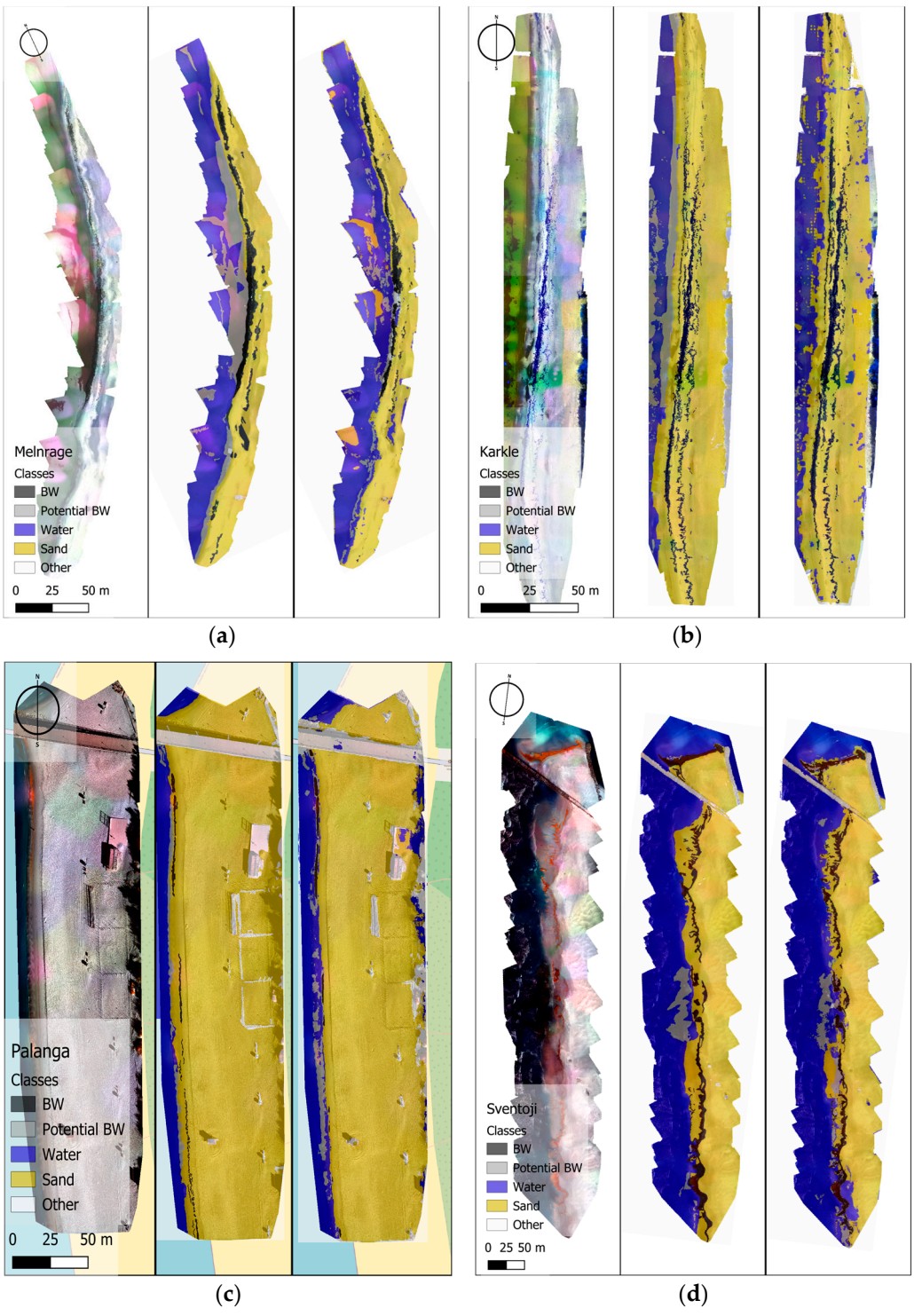

**Figure 10.** Examples of BW spatial distribution in each AOI after UAV image processing with the U-Net model using the "RGB" combination. RGB (left), labeled BW (middle), and modeled BW (right) maps are provided for (**a**) 16 September 2021 in Melnrage, (**b**) 17 September 2021 in Karkle, (**c**) 15 September 2021 in Palanga, and (**d**) 1 October 2021 in Sventoji. The colors of BW in Sventoji and Karkle are different because they are combinations of green, blue, and NIR bands, making them easier to distinguish visually. The different colors near and above the water are noise (see Section 4.2).

## 4. Discussion

### 4.1. Assessment of U-Net Model Performance in BW Segmentation

The U-Net CNN model exhibited commendable results in BW segmentation, particularly when utilizing the "RGB" combination. The segmentation accuracy not only allowed the delineation of BW but also enabled the estimation of its total area across the selected AOIs, ranging from 8.83 m$^2$ to 3710.01 m$^2$. This capability to accurately segment and subsequently estimate the BW area reaffirms the efficiency of U-Net models in semantic segmentation tasks, especially for high-resolution remote sensing images [52].

To the best of the authors' knowledge, only two studies [15,20] were carried out in the context of UAV monitoring of BW. Both of them performed object-based image analysis (OBIA) and achieved relatively high accuracy (producer accuracy > 80%) in classification. In contrast, our research primarily employed the IoU metric, which is suggested as a superior method, especially when combined with other measures like the F1 score. It is also more reliable as it takes into account the whole area rather than a random sample of points or polygons [53], achieving more reliable ML model performance evaluation. However, the labeling process is time-consuming to achieve metrics that include an entire image, especially for large datasets, as in this study (29 mosaiced orthophotos), but after the first training, the U-Net model can be run on new images and instead of labeling all images, the results can just be adjusted as labels for the new round of training, this way reducing the labeling time and overtime, and improving the model's accuracy and generalizability. While recognizing that the absence of producer accuracy calculations precludes a direct statistical comparison with the referenced OBIA studies, it is suggested that future research should incorporate producer accuracy or equivalent measures to enable such direct comparisons.

Some of the images captured during sunrise featured substantial shadow coverage on the beach due to the westward orientation of the AOIs. Such shadows may influence the CNN model's segmentation precision; however, investigating shadow impacts would entail a controlled experimental design that would distract from the study's core objectives. Future research should factor sun position in to minimize shadow occurrence during UAV imagery collection for BW segmentation. External elements like cloud cover and sun angle significantly impact UAV imagery quality [54]. Moreover, accurately pinpointing the waterline in UAV imagery remains a persistent challenge due to the sea surface's ever-changing nature, as noted by Long et al. [55] and Brouwer et al. [56].

The training duration can be extensive, especially with large datasets and intricate models. In our scenario, with 17 tiles measuring 5000 × 5000 each and more than four encoder layers, the "5 bands and height" took roughly 4 h for 100 epochs. Nonetheless, predicting an individual image tile only takes about 5 min, which is important for management tasks that need to estimate quickly whether the amount of BW should be removed. The processing time is also essential, especially as monitoring scales increase. One way to improve it could be the employment of architectures that merge an anchor-free detector with a region-based CNN, which has demonstrated superior precision and faster inference speeds, which is advantageous for smaller datasets [57].

### 4.2. Model Transferability

In general, the IoU values for BW were consistently moderate using all combinations, suggesting that the model's generalizability and transferability in time are possible, considering that the dataset encompassed images captured during varied seasons and under diverse weather conditions, and ensuring a comprehensive representation, contrary to previously mentioned studies. Such results resonate with the broader understanding that UAVs are potent tools for monitoring diverse beach aspects, from mixed sand and gravel to litter [58,59].

The transferability to unseen AOIs could be complicated, as good results were achieved for three AOIs (Sventoji, Melnrage, and Palanga) with relatively homogenous surfaces, characterized by sedimentological uniformity with minimally varying geomorphic attributes

and objects, ensuring a predictable substrate across the examined terrain. Differently from other AOIs, surface conditions were heterogenous in Karkle, which could explain in the low performance of combinations that included heights (BW IoU = 0.37) compared to other data combinations (BW IoU from 0.39 to 0.56), suggesting that heights acquired using the methods in this study should be used carefully. Additionally, the diminished IoU results after incorporating height in both RGB and multispectral data indicate potential errors in the derived heights, or that an overload of layers might be confounding the model; this aligns with the observations of Pichon et al. [60] and Gruszczyński et al. [61]. The accuracy of height could be improved by taking images with oblique angles in addition to nadir, increasing the information available for DSM calculations using structures from motion algorithms [62].

Additionally, the "augmented data" combination did not exhibit a significant divergence from the "5 bands and height" combination. This observation suggests that the model does not exhibit a bias towards the spatial localization of objects within the image. Consequently, this reinforces the notion of the model's transferability across varied scenarios where objects and areas may be positioned differently within the AOI, indicating the model's adaptability in handling them effectively.

### 4.3. Data Combination Influence on the Results

The model's effectiveness varies with different data combinations and classes. Notably, the "5 bands" combination had decent results for the potential BW segmentation, achieving an F1 score of 0.57 and an IoU of 0.40. However, this was inconsistent across the classes of sand, water, and other. The performance inconsistencies across data combinations, such as the superior results of the "RGB" in the BW class but not universally, signal the need for future exploration. While the IoU results for BW were anticipated to be the best with the "5 bands and height" combination due to its comprehensive data, the outcomes were the opposite (IoU = 0.38), and the "RGB" combination IoU was 0.42; however, the difference between combinations was not significant. This suggests that for the segmentation of chosen classes, simpler sensors (such as RGB cameras) could be employed as the accuracy is not worse than with multispectral ones, and the training and prediction time for fewer bands is also shorter. This finding contradicts other studies that found that for multispectral combinations, segmentation accuracy is improved [63].

In this study, equal weights were used for different bands; however, a potential need for different weight distributions in the initial U-Net model for various bands and classes could improve the results of multispectral combination, as hinted by Amiri et al. [64] and Matuszewski et al. [65]. Therefore, the "RGB" combination's surprising efficacy further stresses the need for model adjustments, such as the depth and complexity of CNN models. Rao et al. [66] noted that deeper models can achieve higher detection accuracies but demand more parameters and longer training and inference times.

Data pre-processing and augmentation are equally impactful on CNN performance. As pointed out by Rodrigues et al. [67], CNNs generally fare better with non-pre-processed images when trained from scratch. Thus, the pre-processing and augmentation approach for various combinations could be responsible for the disparities observed across different classes. Moreover, selecting activation functions and optimization methods can also lead to differentiated results. For example, S. Dubey et al. [68] observed that the diffGrad optimizer excels when training CNNs with varied activation functions.

To find the relative importance of each spectral band in the U-Net model, it is suggested to perform a feature ablation analysis, where bands are individually omitted to observe the effect on segmentation accuracy [69]. Additionally, feature permutation importance could be employed, shuffling band values to quantify their impact on model performance [70]. Furthermore, Grad-CAM could provide insight into which bands most influence the predictions of model through gradient-based importance mapping [71]. These methodologies could enable a precise understanding of each band's role in the model's

functionality. In this study, these techniques were not employed, but it would be beneficial for future work to test these techniques to optimize the model's spectral band selection.

Exploring the U-Net model's synergy with other technologies or data sources could be beneficial. Thomazella et al. [72] documented the efficacy of drone imagery merged with CNNs for environmental monitoring. Given the promising results of the "RGB" and "RGB and height" combinations, integrating them with resources like satellite images could create a more comprehensive system for coastal environment monitoring.

*4.4. Class Influence on the Results*

The model's challenges become particularly discernible in its capacity to detect potential BW. The complexities in detecting this class are largely due to the inherent complexities of aquatic environments and underwater light behavior. A primary challenge stems from how water impacts light absorption and reflection [73], with optical complexities in water bodies rendering some remote sensing algorithms less effective. Light shifting at varying water depths can modify the spectral characteristics of reflected light, affecting the model's capability to accurately segment potential BW. Furthermore, the sun's glint can overshadow the upwelled water-leaving radiance during elevated solar angles. As Gagliardini et al. [74] noted, this leads to noise in the image information. Overstreet and Legleiter [75] further demonstrated that sun glint might induce over-corrections in shallow areas of water in the imagery, producing unreliable data. Factors such as wave activity and sea surface roughness add complexity to the water's optical properties, affecting the quality of remote sensing reflectance, as described by Zhang et al. [76]. Improving the segmentation of potential BW could be achieved by adding further pre-processing steps that would correct for water depth [77] and the sun glint effect [78,79].

The limitation of potential BW detection in shallow coastal waters holds significant implications. The deposition of potential BW, especially in vast amounts under intense heat, requires its prompt removal to uphold the beach's ecological equilibrium, smell, and visual appeal. Overlooked potential BW might lead to significant underestimations of BW deposition on beaches, thereby affecting beach management.

This study recommends prioritizing the use of "RGB" data configurations for U-Net CNN applications in BW segmentation due to their moderate accuracy and lower computational demand. It is recommended to re-evaluate the inclusion of height data from UAVs, as it did not significantly improve and sometimes even reduced model precision. Beach managers should consider these findings to optimize BW monitoring workflows, ensuring that methods remain cost-effective and suitable for various beach types without the need for high-end computing resources. This approach will help in scaling up coastal monitoring efforts while maintaining efficiency and accuracy.

While this study has laid important groundwork in applying U-Net CNN models for BW segmentation using UAV imagery combinations, there remain areas for enhancement. Future studies could benefit from incorporating a wider range of environmental conditions and beach morphologies to strengthen the model's generalizability. Moreover, integrating advanced data pre-processing techniques to reduce the effects of variable water reflectance could further refine segmentation accuracy. Additionally, employing a systematic approach to evaluate the impact of individual spectral bands on the model's performance could provide deeper insights into the model's interpretability and guide more efficient feature selection.

## 5. Conclusions

The U-Net model showed promising results using a model trained only on the "RGB" combination for validation data, where the accuracy of BW segmentation was moderate (IoU = 0.42 and F1 score = 0.54), while a relatively better accuracy (F1 score = 0.92 and IoU = 0.59) was achieved for the overall model (the segmentation of all classes). The achieved segmentation accuracy enabled a consistent estimation of BW across the studied AOIs, and BW was found to be in a range of 8.83 $m^2$ to 3710.01 $m^2$. However, the model

underperformed in the segmentation of potential BW, influenced by the inherent challenges presented by variable water reflectance, which might be modulated by factors such as wave patterns, turbidity, transparency, depth, and sun glint. The empirical evidence confirmed a notable degree of transferability in the deployment of the U-Net model across other locations with similar geomorphology of beaches (e.g., sandy or pebble beaches) to those utilized in the training data.

Contrary to the initial hypothesis, incorporating all spectral bands did not improve the model's performance across all classes. Interestingly, the inclusion of height data, acquired from UAV DSM that were only acquired using nadir-facing images, should be reconsidered as the heights will not have accurate information.

Finally, this study underscores the utilization of U-Net CNNs for BW detection, demonstrating that effective model training and analysis can be conducted without the reliance on high-end computing resources, thereby allowing for more accessible and scalable applications in BW monitoring and management.

**Author Contributions:** E.T.: conceptualization, methodology, software, validation, formal analysis, investigation, resources, data curation, writing—original draft, writing—review and editing, visualization, supervision; M.B.: conceptualization, methodology, validation, investigation, resources, data curation, writing—review and editing, supervision; D.V.: conceptualization, validation, investigation, resources, data curation, writing—review and editing, supervision; J.G.: conceptualization, validation, investigation, resources, writing—review and editing; M.K.: resources, funding acquisition, writing—review and editing, supervision. All authors have read and agreed to the published version of the manuscript.

**Funding:** This research was supported by the Doctorate scholarship program in Ecology and Environmental Sciences at Klaipeda University, Lithuania. The field campaigns were co-funded by the Interreg LAT_LIT Programme, co-financed by the European Regional Development Fund (LLI-525 ESMIC).

**Data Availability Statement:** The data presented in this study are available on request from the corresponding author.

**Acknowledgments:** Authors kindly acknowledge Arūnas Balčiūnas, Greta Gyraitė, Greta Kalvaitienė, and Viktorija Sabaliauskaitė for their support in performing the field work/campaigns.

**Conflicts of Interest:** The authors declare no conflict of interest.

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
