# Peer review of "U-Net Performance for Beach Wrack Segmentation: Effects of UAV Camera Bands, Height Measurements, and Spectral Indices"

_drones, doi:10.3390/drones7110670_

Round 1

Reviewer 1 Report

Comments and Suggestions for Authors

In this research, the authors explored utilizing the U-Net convolutional neural network (CNN) model for the segmentation and monitoring of beach wrack (BW) in coastal environments, employing multi-spectral imagery. They experimented with various input configurations, including "RGB," "RGB and height," "5 bands," "5 bands and height," and "Band ratio indices," to identify the most suitable dataset modification for the U-Net model. Their findings indicated promising results with the "RGB" modification, achieving a moderate IoU of 0.42 for BW and overall accuracy of IoU = 0.59. However, challenges arose in segmenting potential BW due to dynamic lighting conditions in aquatic environments, as factors like sun glint, wave patterns, and turbidity influenced model accuracy. Surprisingly, integrating all spectral bands did not improve model efficacy, and adding height data from UAVs reduced model precision in both RGB and multispectral scenarios. The study confirmed the potential of U-Net CNNs for BW detection, highlighting the suitability of their proposed method for diverse beach geomorphologies, without requiring high-end computing resources, thereby facilitating more accessible applications in coastal monitoring and management.

The paper is well structured and has a logical flow, however, I have some minor concerns that I have highlighted in the attached file.

Author Response

Please see the attachment.
Thank you for your insightful review and valuable suggestions, which have enhanced the clarity and quality of our manuscript.

Reviewer 2 Report

Comments and Suggestions for Authors

This manuscript tests different combinations of input remotely-sensed variables into U-Net to detect beach wrack. Overall, it is a good paper that is well-written and well executed. I highlighted in the points below my minor comments, and also identified some points of discussion that I would like to see added to the discussion.

Title: Change indexes by indices
In the abstract: define IoU
Line 17: influenced
Lines 21-22: I recommend clarifying the last sentence, as it sounds like it says "without ... more accessible applications in coastal monitoring and
Line 31: remove the first "can"
Line 33: remove "can" and change "have" to "has"
Line 58: add "such as" before "aeriel photography"
Line 66: I would suggest changing "conventional" for "satellite-based"
Line 74: Remove "earlier exhibited"
Line 91: It would be relevant to list which 5 bands
Line 105: Change "modifications" by "combinations"
Line 132: Remove the first "the"
Lines 133-134: Is "generally" necessary here?
Line 140: The total length of the four beaches? If so, change "beach" for "beaches"
Lines 143-145: Use the proper nomenclature for taxonomy (e.g., italics)
Lines 158-159: The degrees sign should be made different than an o.
Line 175: Has the RGB camera been described? Up to this point, the reader is just informed of the RedEdge-MX, so it's unclear that there are multiple cameras. First mention of the Zenmuse comes at line 185.
Lines 202-203: I'm not sure to understand the link between the masking and the labeling steps. Why was masking done? Please clarify it in the text.
Line 232: What is the augmentation process? Could it be described earlier? Maybe even in the caption of Figure 2, if deemed appropriate?
Line 233 and elsewhere: I think "modifications" should be "combinations"
Line 236: When the authors say "heights", do they mean DSM, DTM, or both? This should be clarified in a few places. There, too, the readers don't know which indices were used, what augmented is, and that the five bands are those from the RedEdge camera. It should all be clarified or the text should be shuffled  to provide all the information to the readers in the right order.
In the three equations, the authors could remove the letters "Rrs" from all the bands and it would simplify the equations and make them easier to read.
Line 253: Which type of chlorophyll (a, b, c, d, or mulitple ones)?
Line 277: "utilized" for "conducted"?
The labels and text in Figure 5 could be bigger.
How accurate is the measurement of heights with a ruler? How did the authors ensure that the ruler wouldn't penetrate the sand, and where were the measurements taken (i.e., at the highest point of BW within a certain radius)?
Line 339: below 0.5 as low?
Figure 10: The scale bars could all be limited to 50m, saving some space on the figures. The RGB colors are a bit odd in some places. Can the authors comment on this?
Lines 505-510: I understand the point that the authors are making, but not calculating producer accuracy means that they cannot compare their results directly with those other studies. This is problematic as a direct comparison should be made to assess whether their technique works best.
Section 4.1 is not well titled, since there is no direct comparison of results and metrics with other studies.

Missing points of discussion to consider:
Why did Karle perform differently (cf. lines 417-419)?
Is there a way to know which of the three band ratio indices contributed most? Or even the relative contribution of let's say IR or red-edge bands to the models?
Any things that could be improved, for example to better measure heights of BW?
What are the main recommendations the authors would like to make?

Comments on the Quality of English Language

Integrated into the comments above.

Author Response

(The authors gave the same response as above.)
